# Structural Characterization of Foxtail Millet (*Setaria italica*) Polysaccharides and Evaluation of Its Antioxidant and Immunostimulatory Activities

**DOI:** 10.3390/antiox14010113

**Published:** 2025-01-20

**Authors:** Haiying Zhang, Chengyu Peng, Wei Zhang, Huatao Liu, Xiaodong Liu, Changqing Sun, Xiaoning Cao

**Affiliations:** 1College of Agriculture, Shanxi Agricultural University, Taiyuan 030000, China; 2Shanxi Province Key Laboratory of Sustainable Dryland Agriculture, Organic Dryland Agriculture Research Institute, Shanxi Agricultural University, Taiyuan 030000, China; 3Center for Agricultural Genetic Resources Research, Shanxi Agricultural University, Taiyuan 030000, China

**Keywords:** foxtail millet, polysaccharide, structural characterization, antioxidant, immunostimulatory activity, RAW 246.7 cells

## Abstract

A water-soluble polysaccharide from foxtail millet (FM-D1) was isolated and purified through gradient ethanol precipitation and column chromatography. Size-Exclusion Chromatography–Multi-Angle Light Scattering–Refractive Index (SEC-MALLS-RI) and high-performance anion-exchange chromatography (HPAEC) analyses revealed that FM-D1 constitutes a highly purified neutral polysaccharide exclusively composed of glucose as the sugar unit, with a molecular weight of 14.823 kDa. The structural characterization results obtained from gas chromatography–mass spectrometer (GC–MS) and nuclear magnetic resonance spectroscopy (NMR) spectra suggest that FM-D1 primarily consists of a main chain linked by →4)-α-D-Glc*p*-(1→ and minor quantities of →4,6)-α-D-Glc*p*-(1→ to form the main chain, with branching mainly composed of α-D-Glc*p*-(1→ attached to the *O*-6 position of →4,6)-α-D-Glc*p*-(1→ sugar residues. Based on these findings, the antioxidant and immunomodulatory activities of FM-D1 were evaluated in vitro. The results indicated that FM-D1 exhibited moderate 2, 2-diphenyl-1-picrylhydrazyl (DPPH) and 2,2′-Azinobis-(3-ethylbenzthiazoline-6-sulphonate) (ABTS) radical scavenging capacity and total antioxidant capacity (TAOC). Furthermore, FM-D1 stimulated macrophage proliferation and inhibited the production of nitric oxide (NO) and inflammatory factors (TNF-α, IL-1β, and IL-6) in lipopolysaccharide (LPS)-stimulated RAW 246.7 cells. Overall, the findings of this study suggest that foxtail millet holds promise as a potential antioxidant agent and immunologic substance in functional foods.

## 1. Introduction

Foxtail millet (*Setaria italica*) is an annual herbaceous plant within the Poaceae family. As a traditional food in China, its dehusked grains have high nutritional value, constituting a rich source of bioactive compounds such as proteins, fats, amino acids, phenolic compounds, sterols, phytic acid, minerals, vitamins, and other bioactive compounds [1]. These components contribute to its high digestibility and absorbability [2,3]. In traditional Chinese medicine, foxtail millet is believed to possess properties that tonify the spleen and stomach, as well as nourish the kidneys [4]. It is highly valued for its nourishing properties and is used for various purposes, including regulating sleep, nourishing Yin and blood, and addressing other health-related needs [5]. Recent research has revealed that bioactive compounds extracted from foxtail millet exhibit a range of biological activities, such as lowering blood lipids, reducing blood pressure, antioxidant effects, enhancing immunity, and anti-aging properties [6,7,8,9]. Consequently, the bioactive components of foxtail millet are emerging as important functional food ingredients. Despite their relatively small quantities, these compounds exert positive effects on human health.

Polysaccharides, recognized as vital bioactive macromolecules, are ubiquitous in plant, animal, and microbial cellular structures [10,11,12,13]. They consist of long-chain polymers comprising over ten monosaccharides linked by glycosidic bonds [14]. As essential components of cell membrane receptor molecules, polysaccharides participate in cell recognition, information transmission, and various biological functions [15,16,17]. They serve as nonspecific broad-spectrum immunomodulators and crucial biomaterials across multiple life activities. Numerous biological activities of polysaccharides were reported in the literature, including antioxidant properties [18], anti-tumor effects [19], anti-inflammatory effects [20], anti-aging effects [21,22], and immune regulation [23,24,25]. Among these activities, the immunoreactivity and health benefits of macromolecular polysaccharides have garnered widespread attention. Accumulating evidence suggests that polysaccharides extracted from plants can activate RAW264.7 macrophages by producing NO and various anti-inflammatory factors, thereby exhibiting effective immunostimulatory activity [26,27,28]. For example, glucan was proven to be a crucial immunomodulatory polysaccharide [29]. It was demonstrated that a homogeneous glucan extracted from the roots of *Codonopsis pilosula* exhibited the potential to enhance immune activation and contribute to human health through immune regulation [30]. It was also reported that polysaccharides derived from *Russula vinosa* Lindblad could exert immune effects by stimulating RAW264.7 cells to secrete IL-1β and TNF-α cytokines. Polysaccharides represent key functional components in foxtail millet. However, research on the structural characterization and bioactivity assessment of these polysaccharides is limited, primarily focusing on refining extraction techniques [31]. Therefore, we speculated that foxtail millet polysaccharide has antioxidant and immunomodulatory activities. Foxtail millet is an excellent functional food, hence foxtail millet polysaccharides may also be a potential functional food. The functional activities of foxtail millet polysaccharide were closely related to its structure. Hence, it is imperative to investigate the structure of foxtail millet polysaccharides for the development of functional food.

In this study, a glucan polysaccharide was isolated from foxtail millet utilizing a water extraction and alcohol precipitation method. It was further purified via gel permeation chromatography using an AKTA purification system. Subsequently, the structural characteristics were identified using Fourier transform infrared (FT-IR) spectroscopy, a gas chromatography–mass spectrometer (GC–MS), nuclear magnetic resonance spectroscopy (NMR), and a scanning electron microscope (SEM). Furthermore, the in vitro antioxidant activity (DPPH, ABTS, and T-AOC) was assessed. Next, the potential immunostimulatory activity on murine RAW264.7 macrophages was evaluated by quantifying the secretion levels of NO and inflammatory cytokines (TNF-α, IL-1β, and IL-6). The purpose of this study was to provide further insight into the structure and biological activity of foxtail millet polysaccharides, providing a theoretical basis for the development and advancement of functional foxtail millet polysaccharide foods, thereby enhancing the economic value of millet resources.

## 2. Materials and Methods

### 2.1. Materials and Reagents

The “Shennong 2” foxtail millet cultivar was sourced from an experimental site in Dongyang Town, Shanxi Agricultural University, Jinzhong City, Shanxi Province. Prior to processing the samples, a meticulous selection process was carried out to eliminate blighted or damaged grains, as well as any impurities. Subsequently, the grains were dehusked, ground, and pulverized through a 60-mesh sieve. Lastly, they were packaged and stored in a refrigerator for future use.

DEAE-52 Sepharose Fast Flow and Sephacryl S-400 HR were purchased from Lanxiao Technology Co., Ltd. (Xi’an, China) and G.E. Company (Fairfield, CT, USA). Standard monosaccharides, including fucose, rhamnose, arabinose, galactose, glucose, xylose, mannose, fructose, ribose, galacturonic acid, glucuronic acid, mannuronic acid, and guluronic acid were procured from Sigma Co. (St. Louis, MO, USA). Antioxidant test kits, including the DPPH radical scavenging kit, ABTS radical scavenging kit, and T-AOC assay kit, were purchased from Beijing Solarbio Technology Co., Ltd. (Beijing, China). RAW264.7 murine macrophage cells were obtained from Tianjin Chuangke Biotechnology Co., Ltd. (Tianjing, China). Dulbecco’s modified Eagle medium (DMEM), fetal bovine serum (FBS), penicillin, streptomycin, and lipopolysaccharide (LPS) were supplied by Thermo Fisher Technology Co., Ltd. (Waltham, MA, USA). The Counting Kit-8 (CCK8) and NO detection kit were purchased from Beyotime Institute of Biotechnology (Shanghai, China), whereas TNF-α, IL-1β, and IL-6 enzyme-linked immunosorbent assay (ELISA) kits were procured from Shanghai Jining Industrial Co., Ltd. (Shanghai, China). All other reagents used in this study were of analytical grade.

### 2.2. Extraction, Refinement, and Purification of Polysaccharides

The extraction of “Shennong 2” polysaccharides was processed using the method described by Chen with some modifications to form crude water extracts [20]. The starch was precipitated with low concentration ethyl alcohol, and then 200 mg a-amylase (400 u/g) was added after redissolution and stirred at 38 °C for 0.5–1 h until the potassium iodide samples did not change color. Then, the dried crude water extracts were dissolved in water and deproteinized using the Sevage method [32]. Subsequently, a petroleum ether reagent was used to remove fat, while macroporous resin AB-8 was utilized for pigment removal. The solution was dialyzed (3000 Da) against water, concentrated, and freeze-dried to yield crude polysaccharide. The total soluble sugar content and purity of crude polysaccharide were determined by the phenol-sulfuric acid method at 490 nm [33,34].

The sample was loaded onto a DEAE-52 Sepharose Fast Flow column (26 mm × 400 mm) and sequentially eluted with distilled water at 4 mL/min, followed by elution with 0.1, 0.2, and 0.3 M NaCl solutions. Each eluted fraction was collected and assayed for the carbohydrate content. Next, the polysaccharide solution was loaded onto a Sephacryl S-400 HR column (26 mm × 1000 mm) and then subjected to elution with distilled water at a flow rate of 1.0 mL/min. The elution process was monitored using the phenol-sulfuric acid method.

### 2.3. Purity Assessment and Molecular Weight Determinationacce

The UV–vis spectrum of the water solution of the sample (5 mg/mL) was measured on a multifunctional microplate reader (Multiskan GO, Thermo Fisher Scientific, Waltham, MA, USA) within the wavelength range of 200–1000 nm, with a scanning interval of 1 nm. Pure water was used as a blank control under the same conditions.

Size-Exclusion Chromatography–Multi-Angle Light Scattering–Refractive Index (SEC-MALLS-RI) was used to measure the homogeneity and molecular weight of the purified fraction. The weight and number average molecular weight (Mw and Mn) and polydispersity index (Mw/Mn) of the fraction in a 0.1 M NaNO_3_ aqueous solution containing 0.02% NaN_3_ were measured based on Chen and Ma’s methods [35,36].

### 2.4. Monosaccharide Composition Analysis

The monosaccharide compositions were analyzed via high-performance anion-exchange chromatography (HPAEC) on a CarboPac PA-20 anion-exchange column (3 by 150 mm; Dionex) using a pulsed amperometric detector (HPAEC-PAD; Dionex ICS 5000+ system) [30]. Approximately 5 mg of the sample was hydrolyzed with trifluoroacetic acid (2 M) at 121 °C for 2 h in a sealed tube. Next, the sample was dried using nitrogen gas. Methanol was added for washing, followed by blow drying, with the methanol wash process repeated 2–3 times. The resulting residue was dissolved in deionized water and filtered through 0.22 μm microporous filtering film prior to measurement. The chromatographic conditions were as follows: the flow rate was maintained at 0.5 mL/min, with an injection volume of 5 μL. The solvent system consisted of three components: solvent system A (ddH_2_O), solvent system B (0.1 M NaOH), and solvent system C (0.1 M NaOH, 0.2 M NaAc). A gradient program was employed, with the volume ratio of solutions A, B, and C set at 95:5:0 at 0 min, 85:5:10 at 26 min, 85:5:10 at 42 min, 60:0:40 at 42.1 min, 60:40:0 at 52 min, 95:5:0 at 52.1 min, and 95:5:0 at 60 min.

### 2.5. FT-IR Spectroscopy of FM-D1

The FT-IR spectroscopy analysis of the foxtail millet polysaccharide of FM-D1 was conducted following Wang’s method [37], with some modifications. For analysis, a small quantity of the polysaccharide sample was mixed with 200 mg of potassium bromide, compacted into a 1 mm thick sheet, and then subjected to machine detection. The scanning analysis was conducted using the Nicolet iZ-10 FT-IR spectrometer (Thermo Nicolet, Waltham, MA, USA), with an instrument resolution of 4.00 cm^−1^. Notably, the scans were performed within 4000-450 cm^−1^, with 32 scans completed for each measurement.

### 2.6. SEM Analysis of FM-D1

The apparent morphologies of the polysaccharides were visualized using SEM (Zeiss Merlin Compact, Jena, Germany). The samples, coated with a thin layer of gold, were positioned on the substrate, and images were observed under high vacuum conditions at a voltage of 1.0 kV and 100- and 4000-fold magnifications.

### 2.7. Methylation Analysis

The methylation of FM-D1 was carried out following the method described by Ciucanu and Kerek [38], with minor modifications. The polysaccharide sample was first dissolved in DMSO. Next, the solution was methylated in a mixture of DMSO and NaOH. After 30 min, CH_3_I was added for reaction for 1 h. After complete methylation, the permethylated products were hydrolyzed using 2 mol/L TFA at 121 °C for 1.5 h, reduced with NaBD_4_, and acetylated with acetic anhydride for 2.5 h at 100 °C. The acetates were dissolved in chloroform and analyzed with GC–MS using an Agilent 6890A-5975C equipped with an Agilent BPX70 chromatographic column (30 m × 0.25 mm × 0.25 µm, SGE, Victoria, Australia). High purity helium, with a 10:1 split ratio, served as the carrier gas, and an injection volume of 1 μL was employed. Mass spectrometry analysis was initiated at a temperature of 140 °C for 2.0 min, followed by a gradual temperature increase to 230 °C at a rate of 3 °C/min for an additional 3 min. The scan mode employed was SCAN, covering a range of *m*/*z* values from 50 to 350.

### 2.8. NMR Spectroscopy

For NMR spectroscopy, the sample was dissolved in 0.5 mL D_2_O to achieve a final concentration of 40 mg/mL. Then, the 0.5 mL dissolved solution was transferred to an NMR tube; both 1D-NMR (^1^H-NMR and ^13^C-NMR) and 2D-NMR (COSY, NOESY, HMBC, and HSQC) spectra were recorded at 25 °C using a Bruker AVANCE NEO 500M spectrometer system (Bruker, Rheinstetten, Germany) operating at 500 MHz [39,40], with a liquid probe QXI 1H/31P/13C/15 N 5 mm quad resonance inverse detection probe. NMR spectra were calibrated with the HOD peak in deuterated water. The solvent peak typically appeared at approximately 4.71 ppm. The NOESY mixing time was 0.6 s.

### 2.9. Antioxidant Activity of FM-D1 In Vitro

The scavenging ability on DPPH and ABTS radicals of polysaccharide FM-D1, as well as T-AOC, were evaluated using commercial kits provided by Solarbio technology Co., Ltd. (Beijing, China) [36,41]. The free radical scavenging activity of FM-D1 was analyzed and compared at concentrations of 0, 50, 100, and 300 μg/mL, with ascorbic acid (vitamin C, Vc) at 100 μg/mL serving as a positive control. Each experimental group was replicated three times. All experimental procedures strictly adhered to the instructions outlined in the kit manual.

### 2.10. Assays of Immunomodulatory Activity of FM-D1

#### 2.10.1. Cell Culture and Proliferation Activity Assay of RAW264.7

The study utilized mouse mononuclear macrophage leukemia cells (RAW264.7) cultured in DMEM supplemented with 10% heat-inactivated fetal bovine serum (FBS), 1% penicillin (100 U/mL), and streptomycin (100 µg/mL). Cell cultures were maintained at 37 °C in a humidified atmosphere with 5% CO_2_.

The effect of FM-D1 on RAW264.7 cell viability was detected by the CCK8 method [42]. Briefly, logarithmic growth phase cells were seeded at a density of 1 × 10^5^ cells/mL, with 100 μL per well in a 96-well plate. The plates were then incubated in a cell culture incubator at 37 °C with 5% CO_2_ for 2 h to facilitate cell attachment. Following cell attachment, the old culture medium was aspirated. The blank control group received 100 μL of DMEM basal medium (without serum), whereas the experimental groups were treated with various concentration gradients (12.5, 25, 50, 100, 200, 400, 800, and 1600 μg/mL) of FM-D1 solutions prepared in DMEM basal medium. The zero-control group (without seeded cells) was included and underwent all other steps identically. Following treatment, the cells were cultured further at 37 °C with 5% CO_2_ for 24 h. Subsequently, 10 μL of the CCK8 solution was added to each well. After an additional 1 h of incubation, absorbance values were measured at a 450 nm wavelength using a microplate reader. Cell proliferation activity was calculated using the following formula:$$Cell\;proliferation\;activity=\frac{As-Ab}{Ac-Ab}\times 100\%$$
where As represents the optical density (OD) value of the sample; Ab represents the zeroing OD value; and Ac represents the OD value of the blank control group.

#### 2.10.2. Assays for NO Production and Cytokine Secretion

RAW264.7 cells were seeded in a 96-well plate at a density of 5 × 10^5^ cells/100 μL. The positive control group was treated with LPS (1 μg/mL), while the blank control group received the DMEM basal medium. The experimental group was treated with different concentrations (100, 200, 400, and 800 μg/mL) of polysaccharide samples and incubated for 1 h, after which 1 μg/mL of LPS was added. The cells were then incubated at 37 °C for 24 h. Notably, three replicate wells were established for each group. After 24 h of culture, the supernatant of the cell culture was collected in sterile centrifuge tubes for subsequent analysis. The level of NO produced by the RAW264.7 cells was determined using Griess assay [43]. In addition, the secretion of TNF-α, IL-α, and IL-1β was measured using an enzyme-linked immunosorbent assay (ELISA) kit (Nanjing, China) according to the manufacturer’s protocol [44].

### 2.11. Statistical Analysis

Statistical data analysis was conducted using SAS 9.3 software, employing Duncan’s multiple range test. Each experiment was replicated in triplicate. The data were expressed as means ± standard deviations (S.D.). The significance of the difference between the two groups was determined using ANOVA, with a significance level set at a probability value < 0.01 (*p* < 0.01).

## 3. Results

### 3.1. Extraction and Purification Analysis

#### 3.1.1. Ion Purification

The deproteinized, defatted, and decolorized foxtail millet polysaccharides underwent further separation and purification using a weak anion-exchange chromatography column called DEAE Seplife FF. The elution curves of the samples were plotted with the same number of eluent tubers as the abscissa and the total sugar contents (in blue) and the same NaCl concentration in the eluent (in green) as the ordinate, as displayed in Figure 1A. During the ion exchange purification process, three elution peaks of foxtail millet polysaccharides were identified, including one neutral polysaccharide (FM-D1) and two acidic polysaccharides (FM-D2 and FM-D3). The dry sample weights for FM-D1, FM-D2, and FM-D3 were 11,190.2, 200.8, and 55.6 mg, respectively, and the purity of the three polysaccharides was 94.7%, 96.1%, and 54.4%, respectively. Notably, the content of FM-D1 was substantially higher compared to that of FM-D2 and FM-D3, suggesting that FM-D1 constituted the predominant component of water-soluble millet polysaccharides. Therefore, FM-D1 was subjected to enrichment, dialysis, and further purification.

#### 3.1.2. Gel Column Chromatography

The main component of foxtail millet polysaccharide FM-D1, as obtained in Figure 1A, was subjected to Sephacryl S-400HR gel column chromatography for elution. The results are shown in Figure 1B. Gel column chromatography resulted in the separation of FM-D1 into a single elution peak, indicating that the water-soluble polysaccharide FM-D1 from foxtail millet was a highly purified single-component polysaccharide. Following the enrichment and drying of this component, the polysaccharide FM-D1 sample was obtained.

### 3.2. Purity Identification and Molecular Weight Determination

Ultraviolet spectroscopy assessed the purity of the extracted and purified FM-D1. In Figure 2A, it is evident that compared to the blank control, the sample exhibited negligible absorption within the wavelength range of 200–400 nm, suggesting the absence of impurities such as pigments, nucleic acids, and proteins. This indicates a good purification effect. It is well-known that the molecular weight of a polysaccharide is closely related to its biological activity. Therefore, we determined the molecular weight of the foxtail millet polysaccharide FM-D1. As shown in the SEC-MALLS-RI spectrum (Figure 2B and Appendix A), FM-D1 exhibited a symmetric chromatographic peak within the elution time range of 29–32 min. The average molecular weight (Mw) of FM-D1 was determined to be 14.823 kDa, the number average molecular weight (Mn) was 14.619 kDa, the z-average molecular weight (Mz) was 14.972 kDa, and the Mw/Mn was 1.104. As Mw/Mn approaches 1, it signifies improved dispersity, with the molecular weight distribution becoming more concentrated. This observation suggests that FM-D1 is a compound exhibiting relatively uniform characteristics and a more concentrated molecular weight distribution. The monosaccharide analysis results corroborated the UV spectroscopy findings, indicating minimal traces of proteins and starch. This alignment further confirms the high purity of the FM-D1 polysaccharide, rendering it suitable for subsequent chemical structure investigations. Existing research suggests that polysaccharides with low molecular weights possess excellent biological activity, underscoring the need for further exploration in this area [45].

### 3.3. Monosaccharide Analysis

In this study, the HPAEC-PAD method was used to detect the derivatized polysaccharides, and their monosaccharide composition was determined by comparing peak retention times with those of standard substances. The high-performance liquid chromatography analysis results of the mixed standard monosaccharide hydrolysates and FM-D1 are shown in Figure 3A,B. The experiment reveals that FM-D1 comprises solely glucose.

### 3.4. FI-IR Analysis

FT-IR spectroscopy is an indispensable technique in the analysis of polysaccharide structures, which enables the determination of the main functional groups and configurations of sugar ring connections based on their infrared fingerprint region. Notably, infrared light is absorbed when the infrared spectrum of a compound induces a modification in the molecular dipole moment. This absorption is used to attribute and analyze the structure of the sample [46]. The infrared spectrum of FM-D1 is shown in Figure 4, wherein the absorption band observed at 3600–3200 cm^−1^ corresponds to the stretching vibration absorption peak of -OH, characteristic of sugars. The specific details are as follows: the absorption peak at 3413.33 cm^−1^ corresponds to the stretching vibration absorption peak of O-H, while the absorption peak at 2932.00 cm^−1^ is attributed to the C-H stretching vibration. Furthermore, an absorption peak at 1025.03 cm^−1^ is attributed to C-O stretching vibration, whereas the absorption at 847.46 cm^−1^ suggests the presence of α-glycosidic bonds within FM-D1 [47].

### 3.5. Analysis of Glycosidic Bond Types of FM-D1

Methylation analysis can reveal the structural characteristics of polysaccharides, including the types and proportions of glycosidic bonds, and detect monosaccharide units, even in trace amounts. Figure 5 shows the total ion flow diagram associated with the sampling. Table 1 shows the chemical shift distribution of glycosidic bonds in FM-D1, identifying four derivatives, t-Glc(p), 4-Glc(p), 3,4-Glc(p), and 4,6-Glc(p), with relative molar ratios of 14.73:72.16:1.08:12.02. This suggests that 4-Glc(p) is the primary glycosidic bond type in FM-D1. However, further confirmation of the specific structure of FM-D1 is warranted through NMR spectroscopy.

### 3.6. NMR Analysis

In studying polysaccharide biomacromolecules, comprehensive structural insights are primarily obtained through NMR spectroscopy (Figure 6 and Appendix A). The ^1^H NMR and ^13^C NMR spectra of FM-D1 are shown in Figure 6A,B. Figure 6A reveals that the proton spectrum signals are mainly clustered within the δ 3.0 to 5.5 ppm range, with a concentration in the δ 4.3–5.4 ppm region indicative of an anomeric signal region. Multiple coupling signal peaks are identified, suggesting the presence of numerous sugar residues with respective chemical shifts of δ 5.31, 5.27, and 4.87 ppm for the anomeric hydrogens. Non-anomeric hydrogen signals predominantly reside within the δ 3.1~4.2 ppm region, while certain signals require the combination of COSY (Figure 6C) and HSQC (Figure 6D) spectra for the assignment of chemical shifts in H2-H6 for each sugar residue due to significant overlap. The robust signal peak near δ 4.71 ppm corresponds to the solvent peak.

Multiple signal peaks were detected in the anomeric carbon region of the sample. Through the integration of the ^13^C NMR spectrum and the cross-peaks in the anomeric region of the HSQC spectrum, it was ascertained that the anomeric signals present in the sugar were δ 5.31/99.72, 4.87/98.53, and 5.27/99.96 ppm, designated as sugar residues A, B, and C, respectively. In addition, through the integration of the sample’s methylation information, anomeric signals, and extensive literature references, it was inferred that sugar residue A corresponds to →4)-α-D-Glc*p*-(1→, residue B represents α-D-Glc*p*-(1→, and residue C denotes →4,6)-α-D-Glc*p*-(1→ [48].

For residue A, the anomeric signal at δ 5.31/99.72 ppm (H1/C1) indicates a possible α-configuration for the glucose residue. The H2 signal at 3.51 ppm for residue A was corroborated by the COSY spectrum cross-peak observed at δ 5.31/3.51 ppm. Similarly, the sequential determination of signals H3-H6 of the sugar residue A yielded chemical shifts of δ 3.89 ppm, δ 3.57 ppm, δ 3.75 ppm, and δ 3.84 ppm (δ 3.67 ppm), respectively. After assigning the chemical shifts in the hydrogen atoms on the sugar ring, the chemical shifts in C2–C6 on the sugar ring were determined through the HSQC signal, yielding chemical shifts of δ 71.49 ppm, δ 73.31 ppm, δ 76.63 ppm, δ 71.13 ppm, and δ 60.43 ppm, respectively. The lower-field chemical shifts observed for C1 and C4 suggest that this residue bears substitutions at positions *O*-1 and *O*-4 of the sugar ring. Combining the findings from methylation analysis and the existing literature reports, it is inferred that sugar residue A is possibly →4)-α-D-Glc*p*-(1→ [49].

For residue B, the anomeric signal observed at δ 4.87/98.53 ppm (H1/C1) suggests that residue B is likely an α-configured glucose residue. The signals H2-H6 of sugar residue B were sequentially determined using the COSY spectrum, revealing chemical shifts of δ 3.5 ppm, δ 3.67 ppm, δ 3.32 ppm, δ 3.59 ppm, and δ 3.87 ppm (δ 3.75 ppm), respectively. After assigning the chemical shifts in the hydrogen atoms on the sugar ring, the chemical shifts in C2–C6 on the sugar ring were determined through the HSQC signal, yielding chemical shifts of δ 71.71 ppm, δ 72.83 ppm, δ 69.27 ppm, δ 72.67 ppm, and δ 60.38 ppm, respectively. The lower-field chemical shift observed for C1 suggests that this residue has a substitution at position *O*-1 of the sugar ring. Combined with methylation analysis results, it is inferred that sugar residue B is possibly α-D-Glc*p*-(1→ [48,50].

For residue C, the anomeric signal observed at δ 5.27/99.96 ppm (H1/C1) suggests that residue C is possibly an α-configured glucose residue. The H1 chemical shift δ 5.29 ppm of sugar residue C was determined by ^1^H NMR. In addition, the signals H2-H6 of sugar residue C were sequentially determined using the COSY spectrum, yielding chemical shifts of δ 3.53 ppm, δ 3.87 ppm, δ 3.74 ppm, δ 3.53 ppm, and δ 3.83 ppm, respectively. After assigning the chemical shifts of the hydrogen atoms on the sugar ring, the chemical shifts in C2–C6 on the sugar ring can be determined through the HSQC signal, yielding chemical shifts of δ 71.84 ppm, δ 74.48 ppm, δ 76.78 ppm, δ 73.96 ppm, and δ 67.93 ppm, respectively. The lower-field chemical shifts observed for C1, C4, and C6 suggest that this residue has substitutions at positions *O*-1, *O*-4, and *O*-6 of the sugar ring. Combining with the methylation analysis results, it is inferred that sugar residue C is possibly →4,6)-α-D-Glcp-(1→ [15].

The structure and connectivity of the polysaccharide were determined based on the chemical shifts of ^13^C and ^1^H for each sugar residue in FM-D1 (Table 2), combined with the analysis of the HMBC (Figure 6E) and NOESY (Figure 6F) spectra. According to the HMBC spectrum, the connectivity order of each residue in the polysaccharide was inferred: sugar residue A-H1 exhibited cross-peaks at δ 5.31/76.63 ppm with residue A-C4, and at δ 5.31/76.78 ppm with residue C-C4. Sugar residue A-C1 showed a cross-peak at δ 99.72/3.57 ppm with residue A-H4. Sugar residue C-H1 displayed a cross-peak at δ 5.27/76.63 ppm with residue A-C4. Sugar residue C-C1 showed a cross-peak at δ 99.96/3.57 ppm with residue A-H4. Furthermore, aided by the NOESY spectrum, the connectivity order of each residue in the polysaccharide was further inferred as follows: sugar residue A-H1 exhibited cross-peaks at δ 5.31/3.57 ppm with residue A-H4 and at δ 5.31/3.74 ppm with residue C-H4. Sugar residue B-H1 exhibited a cross-peak at δ 4.87/3.83 ppm with residue C-H6. Sugar residue C-H1 showed a cross-peak at δ 5.27/3.57 ppm with residue A-H4.

Therefore, by integrating one-dimensional and two-dimensional nuclear magnetic resonance along with the methylation analysis results, it can be inferred that FM-D1 primarily consists of interconnected units, including →4)-α-D-Glc*p*-(1→ with a smaller proportion of →4,6)-α-D-Glc*p*-(1→, forming the main chain. The branches are mainly composed of α-D-Glc*p*-(1→ units connected to the O-6 position of the sugar residue →4,6)-α-D-Glc*p*-(1→. Consequently, the proposed structure of this polysaccharide chain is illustrated in Figure 7.

### 3.7. Surface Feature Analysis

The composition of monosaccharides, types of glycosidic bonds, and connection methods can significantly influence the microstructure of polysaccharides. Scanning electron microscopy can visually reflect the structure and morphology of polysaccharides, including the size, molecular shape, and porosity [46]. Figure 8A–C shows the surface morphology characteristics of FM-D1. Under a magnification of 5K×, irregular fragmented structures are observed in the polysaccharide sample, with some spherical structures attached to the sheet-like structures, rendering them susceptible to stacking and aggregation.

### 3.8. Antioxidant Activity of FM-D1

#### 3.8.1. DPPH Radical Scavenging Activity

The scavenging effect of polysaccharide FM-D1 against DPPH is shown in Figure 9A. It was found that within the concentration range of 0–300 μg/mL, the DPPH scavenging activity of FM-D1 exhibited an initial increase followed by a subsequent decrease with increasing concentration. At equivalent concentrations, Vc exhibited a higher rate of DPPH scavenging compared to FM-D1. The highest DPPH radical scavenging ability was observed at 50 μg/mL.

#### 3.8.2. ABTS Radical Scavenging Activity

The scavenging effect of polysaccharide FM-D1 on ABTS is illustrated in Figure 9B. Overall, the scavenging activity of FM-D1 on ABTS was lower than that of Vc. Within the concentration range of 0–300 μg/mL, the ability of FM-D1 to scavenge ABTS exhibited a pattern of initially increasing, followed by decreasing, and subsequently increasing again with the increase in FM-D1 concentration. At a concentration of 100 μg/mL, the scavenging effects of FM-D1 and Vc on ABTS were 4.60% and 87.17%, respectively. At 300 μg/mL, FM-D1 exhibits its highest ABTS radical scavenging ability, reaching 6.42%.

#### 3.8.3. Total Antioxidant Capacity

The T-AOC of FM-D1 is shown in Figure 9C. It is evident that within the concentration range of 0–300 μg/mL, the antioxidant capacity of FM-D1 exhibits a trend of initially increasing, followed by decreasing and eventually leveling off with the rise in concentration. At a concentration of 50 μg/mL, FM-D1 demonstrates a maximum antioxidant capacity of 0.1173 μmol/mL. At 100 μg/mL, the T-AOC of Vc was 22.65 times that of FM-D1.

### 3.9. Evaluation of Immunomodulatory Activity

#### 3.9.1. Effect of FM-D1 on Activity of RAW264.7 Cells

After 24 h of treatment with FM-D1, a significant promotion (*p* < 0.01) or no significant effect (*p* > 0.05) on RAW264.7 murine macrophages was observed at concentrations of 100, 200, 400, 800, and 1600 μg/mL compared to the control group (Figure 10). This suggests that at these concentrations, FM-D1 can enhance the activity of RAW264.7 cells without inducing toxic effects. The activation of macrophages is characterized by heightened phagocytic activity, which serves as the frontline defense in the immune response. As is consistent with this finding, Zhang et al. previously reported that a-(1→4) glucan polysaccharide WPMP-1 extracted from *Polygonum multiflorum* exhibits a significant activation effect on the proliferation of splenocytes and peritoneal macrophages compared to the control group [51].

#### 3.9.2. Effect of FM-D1 on NO Production of RAW264. 7 Cells

Figure 11A depicts the impact of FM-D1 on NO production in LPS-induced RAW264.7 cells. NO secretion in the LPS group was significantly elevated compared to the control group, indicating the successful establishment of the LPS model. After treatment with FM-D1 at concentrations of 100, 200, and 400 μg/mL, the NO content was significantly decreased (*p* < 0.01). Furthermore, the NO content progressively increased with the increase in FM-D1 concentration. At 100 and 400 μg/mL concentrations, the NO content reaches the minimum and maximum values of 0.98 and 7.72 μΜ/mL, respectively.

#### 3.9.3. Effect of FM-D1 on Secretion of TNF-α, IL-1β, and IL-6 from RAW264.7 Cells

The effects of FM-D1 on the production of TNF-α, IL-1β, and IL-6 in LPS-induced RAW264.7 cells are shown in Figure 11B–D. It can be seen that after adding LPS, the levels of TNF-α, IL-1β, and IL-6 in the cell culture supernatant were significantly higher compared to those in the control group (*p* < 0.01), indicating the successful establishment of the LPS model where LPS can induce excessive inflammation in cells. Following treatment with various concentrations of FM-D1, the secretion levels of TNF-α and IL-6 significantly decreased (*p* < 0.01). Among them, the low concentration of 100 ug/mL treatment has the most obvious reduction. As for IL-1β, although the decrease in secretion levels was not significant at concentrations of 400 μg/mL, a significant decreasing trend was observed at concentrations of 100 (*p* < 0.05) and 200 μg/mL (*p* < 0.01).

## 4. Discussion

Polysaccharides are a kind of biological macromolecule with high molecular weights and complex branch chains, which are formed by a variety of monosaccharides connected by glycosidic bonds. The biological activity of polysaccharides are closely related to their structure, and the structure mainly depends on the molecular weight distribution, monosaccharide composition, and glycoside linkage [52,53]. In this study, we have analyzed the structure of foxtail millet polysaccharide FM-D1 and found that FM-D1 is composed of a single glucose with a molecular weight of 14.823 kDa. Similar results were reported in other studies. For example, Zhu et al. isolated a homopolysaccharide LPS1 composed of glucose from Longyan [54]. Zhang et al. extracted a polysaccharide detonated as WRP-1 from *Russula vinosa* Lindblad, characterized as a branched β-(1→3)-glucan [30]. Moreover, Cao et al. found that crude polysaccharides extracted from finger millet were predominantly glucose, constituting approximately 88.2% of the total composition [55]. Various studies have also noted differences in the monosaccharide composition and ratios of polysaccharides extracted from the same species, which may be attributed to factors such as the source of the materials, genetic background, and conditions employed during extraction and purification [56]. The structure obtained from the GC–MS and NMR spectra exhibited the backbone of →4)-α-D-Glcp-(1→ and minor quantities of →4,6)-α-D-Glcp-(1→ and the branched chains ofα-D-Glcp-(1→ attached to the O-6 position of→4,6)-α-D-Glcp-(1→ sugar residues. Similar results were reported by other scholars. Li et al. reported a glucan extracted from the root of *Codonopsis pilosula* containing 1-linked β-D-glucose, 1,4-linked α-D-glucose, and 1,4,6-linked α-D-glucose [29]. Zhang et al. reported a glucan composed of 1,4-linked α-D-Glcp, 1,4,6-linked α-D-Glcp, and terminal α-D-Glcp from *Polygonum multiflorum* [51]. Jiang et al. and Niu et al. also reported a similar type of glucan [57,58].

Reactive oxygen species (ROS) are chemical molecules generated during normal metabolic processes within the human body. Oxidative stress and subsequent oxidative damage induced by ROS can contribute to various organ diseases and aging, thereby emerging as significant factors in pathogenesis [59]. Therefore, maintaining a delicate equilibrium between the antioxidant defense system and ROS generation is crucial for safeguarding human health [60]. Given the prevalence of various diseases, antioxidants are sought as a strategic intervention for effectively preventing and treating these conditions. To explore the antioxidant activity of foxtail millet polysaccharides and their derivatives, this study conducted a series of in vitro activity tests, including DPPH and ABTS radical scavenging abilities and T-AOC assays. In this study, it was found that with the increase in the concentration of FM-D1, the free radical scavenging ability and total antioxidant capacity showed a trend of first increasing and then decreasing. These results suggest that the polysaccharides derived from foxtail millet exhibit antioxidant activity. However, their scavenging ability is moderate when compared to Vc, which was consistent with findings reported in other plant polysaccharides [61]. In a word, these results imply that FM-D1 possess promising potential in terms of antioxidant capacity.

Inflammation is an innate defense mechanism of the body triggered by infection, tissue damage, and various other factors. Under normal circumstances, inflammation eradicates harmful stimuli and restores the normal function of injured tissue. However, excessive inflammation can disrupt the homeostasis of normal tissue, and, in severe instances, it may lead to death [52]. Hence, it is imperative to suppress maladaptive or excessive inflammation to maintain human health. Macrophages play critical roles in both innate immunity and cell-mediated immunity. LPS stimulates macrophage activation, leading to the excessive expression of various cytokines and other inflammatory factors, ultimately initiating an inflammatory response. Among them, NO is an essential inflammatory factor in immune regulation processes, regulating cell apoptosis and acting as a defense mechanism against microorganisms and tumor cells within the host [62]. In the current study, it was found that FM-D1 significantly inhibited the secretion of NO by RAW264.7 cells induced by LPS, and the amount of NO produced was concentration-dependent in the concentration range of 100–400 μg/mL. Moreover, the inhibition effect was more obvious at low concentrations. This result suggests that FM-D1 plays an anti-inflammatory role by inhibiting NO secretion at non-cytotoxic concentrations.

In addition to up-regulating the release of NO, the activation of macrophages up-regulates various cytokines, thereby regulating immune and inflammatory responses [63]. Among them, TNF-α, IL-1β, and IL-6 are three typical pro-inflammatory cytokines that play essential roles in innate and adaptive immunity and are critical markers for assessing the degree of inflammation [64]. TNF-α promotes tumor cell death by killing T cells or other cells, and IL-1β and IL-6 play important roles in immune response and protein synthesis during the acute phase [65]. In this experiment, FM-D1 treatment significantly decreased the levels of TNF-α and IL-6 in the test concentration range (*p* < 0.05), and as for IL-1β, FM-D1 significantly decreased its secretion level at low concentrations (*p* < 0.01). These results suggest that FM-D1 can somewhat alleviate inflammation damage by reducing the levels of pro-inflammatory cytokines, demonstrating promising anti-inflammatory activity. However, for the anti-inflammatory cytokine such as IL-10, which is a negative regulator of cell-mediated immune response, we did not do the research. In the next experiment, we will investigate further in detail.

## 5. Conclusions

In this study, a neutral polysaccharide extracted from foxtail millet (FM-D1) underwent separation, purification, and characterization of its basic properties and structures. It was found that FM-D1 is a highly purified polysaccharide composed entirely of glucose as the sugar unit, with a molecular weight of 14.823 kDa. The results showed that FM-D1 is primarily composed of →4)-α-D-Glcp-(1→ and small amounts of →4,6)-α-D-Glcp-(1→ to form the main chain. Additionally, the antioxidant and immune activities of FM-D1 and its derivatives were explored in vitro. Results revealed that FM-D1 exhibited moderate antioxidant activities in terms of DPPH and ABTS radical scavenging capacity and total antioxidant capacity. Moreover, FM-D1 exhibited certain inhibitory effects in the inflammatory response caused by reducing the secretion of NO and inflammatory factors (TNF-α, IL-1β, and IL-6). This study laid a theoretical foundation for further exploration into the structure of foxtail millet polysaccharides and the development of functional foods or medicines.

## Figures and Tables

**Figure 1 antioxidants-14-00113-f001:**
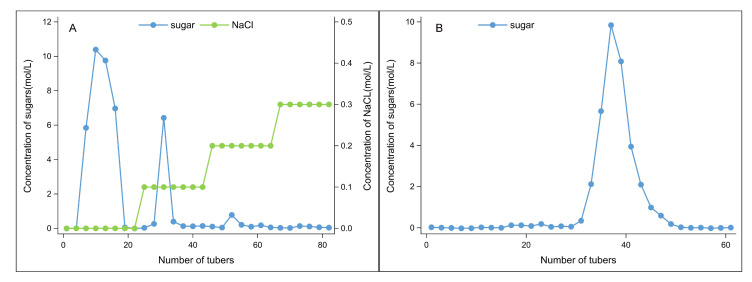
Elution curve of foxtail millet polysaccharide on DEAE FF column (**A**) and Sephacryl S-400HR column (**B**).

**Figure 2 antioxidants-14-00113-f002:**
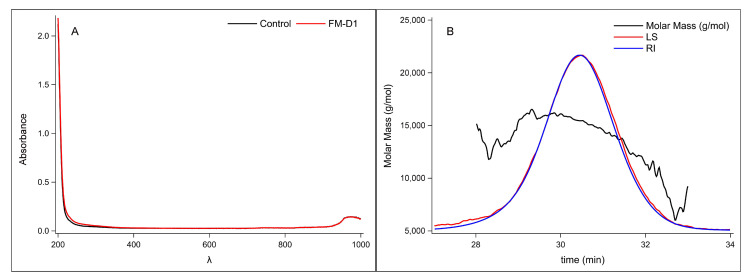
UV–vis spectrum of FM-D1 (**A**) and SEC-MALLS-RI spectrum of FM-D1 (**B**).

**Figure 3 antioxidants-14-00113-f003:**
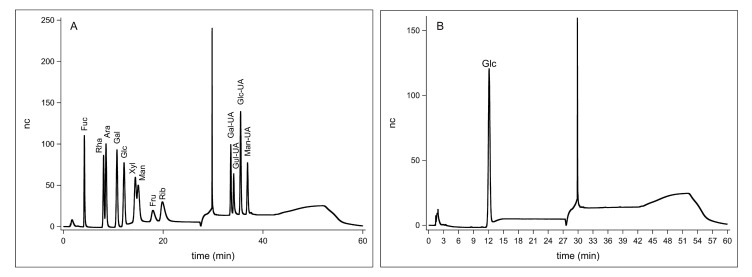
Ion chromatogram of standards (**A**) and ion chromatogram of FM-D1 (**B**). Monosaccharide standards from left to right are fucose, rhamnose, arabinose, galactose, glucose, xylose, mannose, fructose, ribose, galacturonic acid, guluronic acid, glucuronic acid, and mannuronic acid.

**Figure 4 antioxidants-14-00113-f004:**
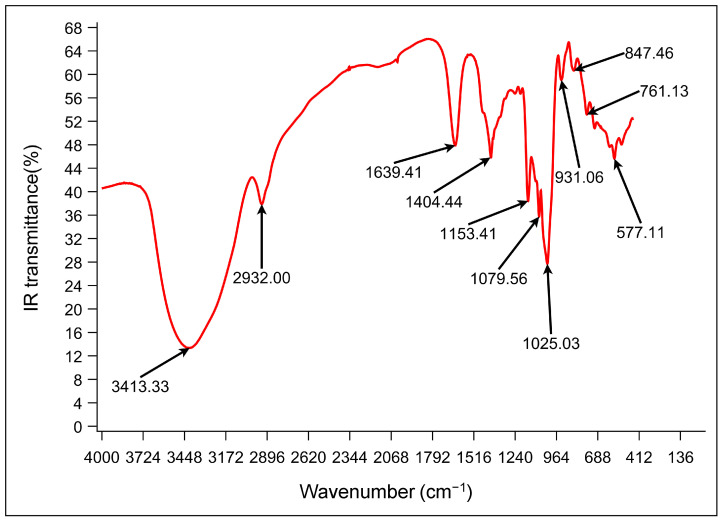
FT-IR spectrum of FM-D1.

**Figure 5 antioxidants-14-00113-f005:**
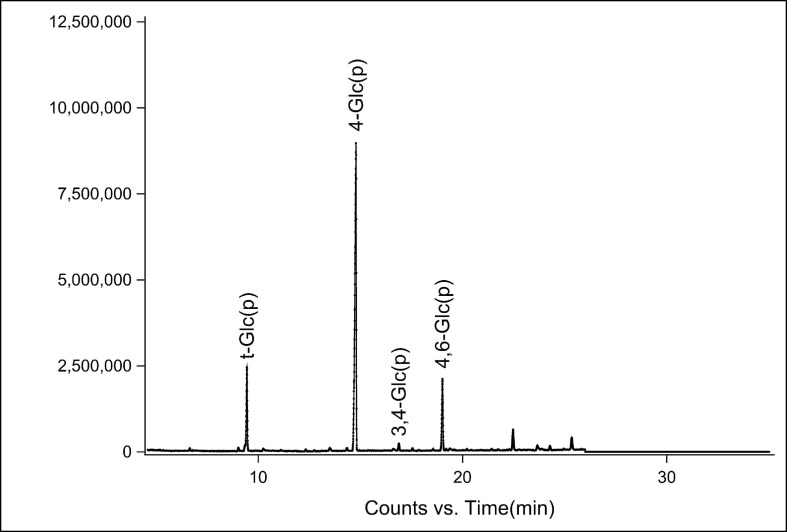
GC-MS total ion flow of purified component FM-D1.

**Figure 6 antioxidants-14-00113-f006:**
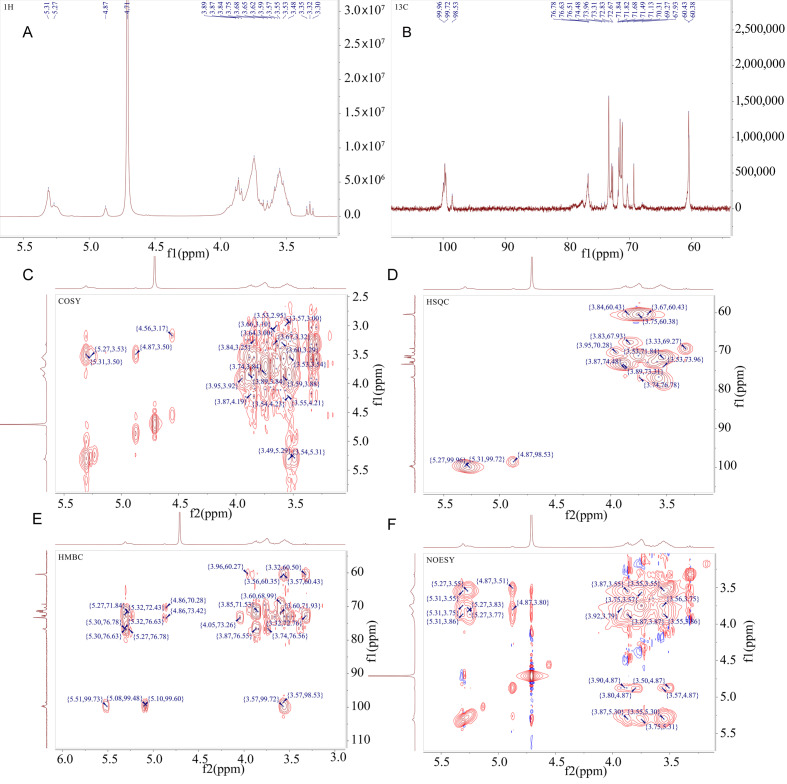
NMR spectral analysis of FM-D1 from foxtail millet (**A**) ^1^H-NMR; (**B**) ^13^C-NMR; (**C**) COSY; (**D**) HSQC; (**E**) HMBC; (**F**) NOESY.

**Figure 7 antioxidants-14-00113-f007:**
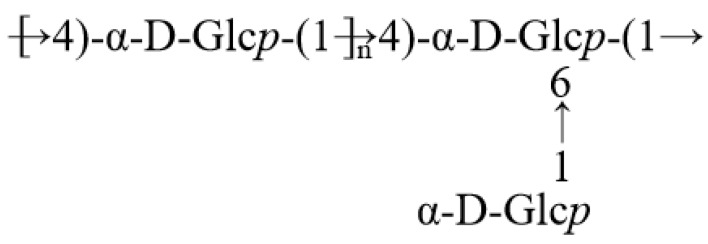
The predicted structure of FM-D1.

**Figure 8 antioxidants-14-00113-f008:**
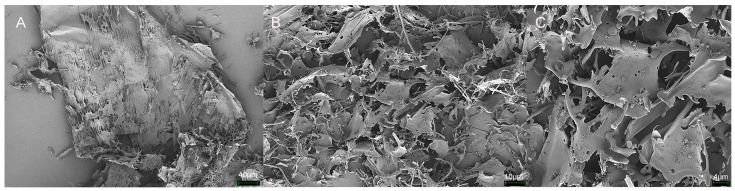
Microstructure observation of FM-D1: (**A**) 500× magnification; (**B**) 2000× magnification; (**C**) 5000× magnification.

**Figure 9 antioxidants-14-00113-f009:**
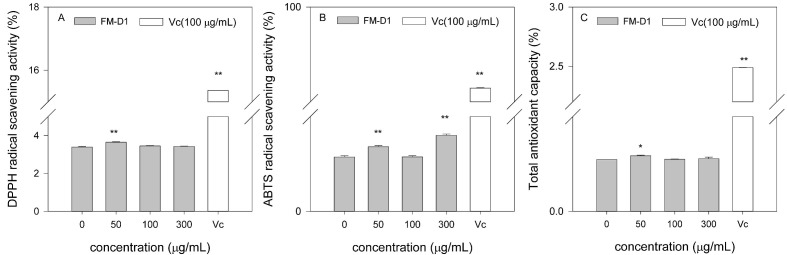
Antioxidant activity in vitro of foxtail millet polysaccharides FM-D1. (**A**) DPPH radical scavenging activity; (**B**) ABTS radical scavenging activity; (**C**) Total antioxidant capacity. * *p* < 0.05 vs. control; ** *p* < 0.01 vs. 0 μg/mL group.

**Figure 10 antioxidants-14-00113-f010:**
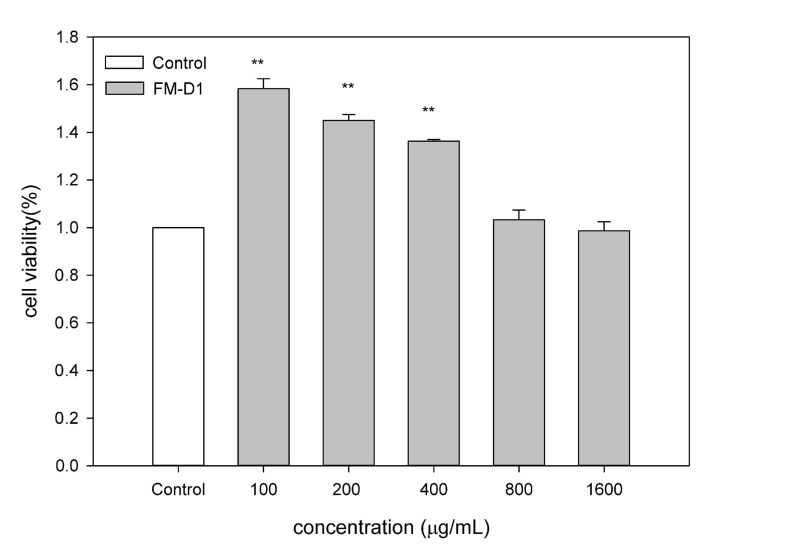
Effects of FM-D1 at different concentrations on cell viability. ** *p* < 0.01 vs. control.

**Figure 11 antioxidants-14-00113-f011:**
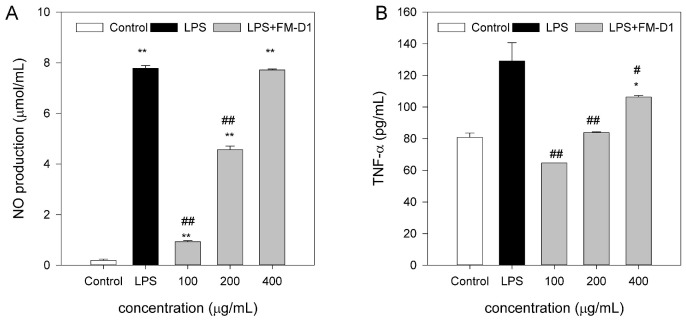
Effects of FM-D1 at different concentrations on release of NO (**A**) and stimulations of TNF-α, IL-1β, and IL-6 (**B**–**D**). * *p* < 0.05 vs. control; ** *p* < 0.01 vs. control; # *p* < 0.05 vs. LPS; ## *p* < 0.01 vs. LPS.

**Table 1 antioxidants-14-00113-t001:** Bonding structure analysis of FM-D1 based on methylation analysis.

Number	Linkage Pattern	Derivative Name	Mass Fragments (*m*/*z*)	Retention Times (min)	Molecular Weight (Mw)	Molar Ratio (%)
1	t-Glc(p)	1,5-di-O-acetyl-2,3,4,6-tetra-O-methyl glucitol	87, 102, 118, 129, 145, 161, 162, 205	9.450	323	14.73
2	4-Glc(p)	1,4,5-tri-O-acetyl-2,3,6-tri-O-methyl glucitol	87, 102, 113, 118, 129, 162, 233	14.786	351	72.16
3	3,4-Glc(p)	1,3,4,5-tetra-O-acetyl-2,6-di-O-methyl glucitol	87, 118, 129, 143, 185, 203, 305	16.886	379	1.08
4	4,6-Glc(p)	1,4,5,6-tetra-O-acetyl-2,3-di-O-methyl glucitol	85, 102, 118, 127, 159, 162, 201, 261	19.017	379	12.02

**Table 2 antioxidants-14-00113-t002:** Chemical shift in glycosyl residues 1H and 13C of FM-D1.

Code	Glycosyl Residues	Chemical Shifts (ppm)					
		H1/C1	H2/C2	H3/C3	H4/C4	H5/C5	H6/C6
A	→4)-α-D-Glc*p*-(1→	5.31	3.51	3.89	3.57	3.75	3.84, 3.67
		99.72	71.49	73.31	76.63	71.13	60.43
B	α-D-Glc*p*-(1→	4.87	3.5	3.67	3.32	3.59	3.87, 3.75
		98.53	71.71	72.83	69.27	72.67	60.38
C	→4,6)-α-D-Glc*p*-(1→	5.27	3.53	3.87	3.74	3.53	3.83
		99.96	71.84	74.48	76.78	73.96	67.93

## Data Availability

The data generated and/or analyzed during the current study are available from the corresponding author on reasonable request.

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
