# Peer review of "Structural Characterization of Foxtail Millet (Setaria italica) Polysaccharides and Evaluation of Its Antioxidant and Immunostimulatory Activities"

_antioxidants, 2025, doi:10.3390/antiox14010113_

Round 1
Reviewer 1 Report
The manuscript presents the interesting topic of isolation and purification of water-soluble polysaccharide from foxtail millet (FM-D1), its structural characterization and then examined antioxidant and immunosuppressive properties. The manuscript is well prepared and contains a lot of experimental data. The structure of the tested substance is documented correctly, but I believe that a Supplement should be prepared for the manuscript in which the authors should include the full ranges of 1H NMR and 13C NMR spectra, as well as 2D NMR correlation spectra and IR , as well as chromatograms and MS spectra. Additionally, in the materials and methods section, please describe in detail how to prepare the sample for NMR tests and provide the technical data of the Bruker spectrometer, because the current data is not sufficient. Making these corrections is necessary.
With such detailed analyses, a Scheme should be created presenting the structures of the compounds that the authors included in Table 1.
Reviewer 2 Report
Nowadays, the topic of polysaccharides is very much in the news, and more and more is being told about their potential beneficial effects on human health. In addition, in the era of current scientific research, evaluation of antioxidant properties alone is not enough, and the authors additionally conducted an analysis of immunostimulatory properties which is very important. Taking into account the whole, that is, the planning of the experiment, the analytical methods used and the results obtained, I believe that the manuscript meets the requirements of Antioxidants and can be considered for publication.
Nevertheless, I have a few issues that need improvement in my opinion:
- in my opinion, with such a small number of samples, a non-parametric analysis should be performed instead of ANOVA analysis
- the keywords must not coincide with the title
- many abbreviations are present in the abstract, all of which need to be explained
- introduction, here I ask for clear wording:
= what is the novelty of this research
= what is the research problem
= what is the purpose of the research
= please formulate the research hypothesis
- methodology, please describe in detail all the methods used; currently this is done in a very abbreviated and sometimes not very clear way
- I believe that the results could be better discussed, please provide a broader analysis with the already available literature in the subject area
- Conclusions, please refer to the research hypothesis you have stated
Round 2
Reviewer 1 Report
The article has been corrected and improved and can be submitted for printing in this version.
The article has been corrected and improved.
Reviewer 2 Report
Accept in current form
Accept in current form